# Reason-and-Execute Prompting: Enhancing MultiModal Large Language Models for Solving Geometry Questions

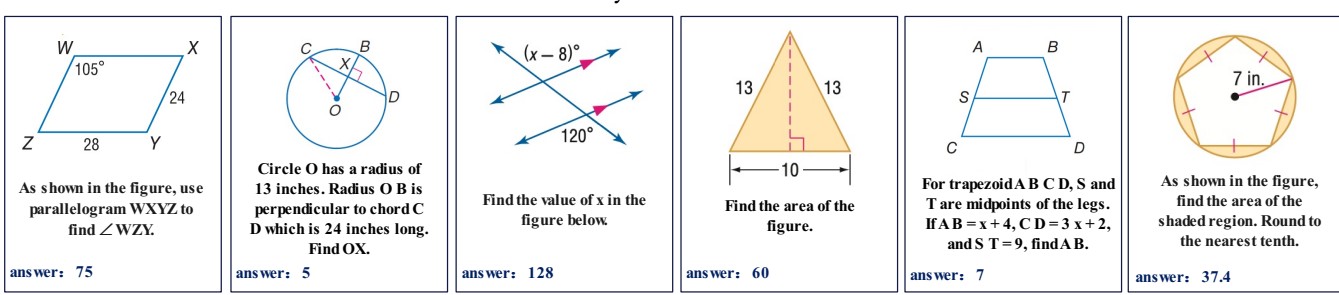

**Figure 1: Examples of answering various geometry questions using Reason-and-Execute Prompting templates.**

## ABSTRACT

MultiModal Large Language Models (MM-LLMs) have demonstrated exceptional reasoning abilities in various visual question-answering tasks. However, they encounter significant challenges when answering geometry questions. These challenges arise due to the need to engage in rigorous reasoning and executing precise arithmetic. To enhance the ability of LLMs to solve multimodal geometric questions, we propose Reason-and-Execute (RaE) prompting: a new prompting method specifically designed for enhancing MM-LLMs to solve geometric questions. Specifically, we first designed a rigorous reasoning process based on domain knowledge of geometry, using a reverse thinking approach, and obtained the precise arithmetic steps required for solving the question. Secondly, based on the analysis of the reasoning process, we designed code blocks in a programming language to implement the arithmetic functions. Finally, by executing the contents of the code blocks using an interpreter, we obtained the answers to the geometric questions. We evaluated the accuracy of 9 models in answering questions on 6 datasets (including four geometry datasets and two science datasets) using different prompting templates. Specifically, in the main experimental result, our RaE showed a maximum enhancement of 12.8% compared to other prompting methods, which proves strong reasoning and arithmetic abilities in solving geometric questions of our method. Moreover, we analyzed the impact of answering from the perspective of solving geometric problems by considering multiple factors, including domain knowledge, geometry shapes, understanding of the question text, and language. This once again emphasizes that our method has passed the comprehensive test of solving geometry questions. The source code and data will be published in a GitHub repository.

**Unpublished working draft. Not for distribution.**

## CCS CONCEPTS

• **Computing methodologies** → **Natural language processing;** **Computer vision; Machine learning algorithms.**.

## KEYWORDS

multimodal large language models, geometry questions, prompting method

## 1 INTRODUCTION

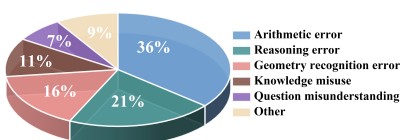

**Figure 2: Error answer analysis of 700 geometry questions with GPT-4V (gpt-4-vision-preview ).**

Traditional methods for solving geometry questions typically focus on mathematical expression[28], while the MultiModal Large Language Models (MM-LLMs) can better understand the relationship between natural language descriptions and geometry shapes [10], as shown in Figure 1. Enhancing the efficiency and accuracy of MM-LLM in solving geometry questions is of great significance for the development of education and intelligent aided systems [18].

The MM-LLMs have demonstrated exceptional reasoning abilities in various visual question-answering tasks[42]. However, there are obstacles in the task of automatically solving geometry questions [22, 28], as shown in Figure 2. The main obstacles include errors in arithmetic results (36%) and errors in logical reasoning processes (21%). At present, there are mainly two methods to overcome these obstacles. One is a fine-tuning method [34, 36, 44] based on specific downstream tasks. Although some MM LLMs are already open source and can be fine-tuned based on pre-trained models [2, 21, 35, 40], they have extremely high requirements for datasets and devices, making it difficult to perform task fine-tuning. Especially in the task of solving geometry questions, firstly, the scale of publicly available high-quality geometry datasets is limited;

Secondly, MM-LLMs typically have billions to tens of billions of parameters, which makes the fine-tuning process require a significant amount of computing resources and time. Compared to this method, another method that uses some examples as prompts [1, 17, 43] to solve new questions is easier to implement and achieves impressive results. Among them, the most representative works are Chain-of-Thought Prompt (CoT)[38] method and Program-Aided Language (PAL) [35] Models. While these methods have demonstrated remarkable performance in various tasks, such as the CoT and PAL prompt methods achieving accuracy rates of 94.7% and 99.2% in mathematical natural language reasoning tasks [35], solving geometry questions remains a significant challenge. For example, Figure 3(a) illustrates that the CoT prompting method, when applied to geometry questions, often misuses data in the reasoning process due to the complexity of domain knowledge, leading to incorrect answers. Similarly, Figure 3(b) shows that the code block generated by the PAL prompting method contains excessive reasoning processes, rendering the program non-executable and unable to provide the answer. Thus, designing a prompting method that can facilitate rigorous reasoning and precise arithmetic for solving geometry questions remains a huge challenge.

To address these challenges, we propose Reason-and-Execute (RaE) prompting: a new prompting method specifically designed for enhancing MM-LLMs to solve geometry questions. Specifically, we first designed a rigorous reasoning process based on domain knowledge of geometry, using reverse thinking [9] approach, and obtained the precise arithmetic steps required for solving the question. Secondly, based on the analysis of the reasoning process, we designed code blocks in a programming language to implement precise arithmetic functions. Finally, by executing the contents of the code blocks using an interpreter, we obtained the answers to the geometric questions.

In the analysis of experimental results, we have demonstrated the ability of the RaE prompting method to perform rigorous reasoning and precise arithmetic operations. In addition, we also analyzed the impact of domain knowledge, geometry shapes, understanding of the question text, and language on our prompt templates for solving geometry questions. We conclude that rigorous reasoning and precise arithmetic processes are essential for accurately solving geometry questions. The contributions of this paper are as follows:

- We propose Reason-and-Execute (RaE) prompting: the first prompting method specifically designed for enhancing MM-LLMs to solve geometric questions.
- We have designed a new prompt template that combines rigorous reasoning with precise arithmetic.
- We analyzed geometric problems from different perspectives and tested RaE prompting method, ultimately achieving impressive results.

## 2 RELATED WORKS

### 2.1 MultiModal Large Language Models

With the multimodal large language models (MM-LLMs) showing a strong ability of image-text understanding [42], the research of math reasoning using MM-LLM combined with images and texts began to appear [22, 24, 25]. Especially in the study of geometric question solutions that often appear in multimodal forms [28],

there has been greater vitality [22]. Specifically, solving geometry questions requires a combination of image and text information to complete professional domain-knowledge reasoning and precise arithmetic operations. Although this is a huge challenge for LLMs [45], the MM-LLMs can fully leverage its advantages [30]. For example, GPT-4(Vision)[29]uses a visual encoder with pre-trained components for visual perception, aligning the encoded visual features with the language model, thereby achieving a comprehensive understanding of geometric problem images and text information; Qwen VL [2] is a large-scale visual language model launched by Alibaba [41] Cloud that performs well in tasks such as image description, question answering, visual positioning, and flexible interaction, moreover the baseline model used in our experiment is its two important models: qwen-vl-chat and qwen-vl-plus; CogVLM [35] puts visual understanding as a higher priority to achieving the deep fusion of visual language features; mPLUG Owl [39] can learn the parameters of the visual encoder in the first stage of training, to achieve efficient image alignment with this article; InternLM-XComposer2 [8] proposes a new fine-tuning method of visual and text alignment, which enhances the visual understanding ability of the model; Yi Vision Language (Yi-VL) [40] demonstrates its strong capabilities in complex interdisciplinary tasks with its excellent ability to understand images and generate dialogue. DeepSeek-VL [21] is an innovative open-source visual language model that stands out for its ability to understand real-world scenarios in various applications such as logic diagrams, web pages, and natural images; Gemini [18] is a large language model released by Google [3], and designed specifically for "general-purpose tasks", Gemini Pro, has performed well in various multimodal processing fields. They have the potential to solve geometry questions by combining images and text.

### 2.2 Prompting Methods

LLMs have achieved tremendous success with the support of computing power and datasets. Computational power enables the model to be sufficiently large, while also possessing excellent comprehension, memory, reasoning, and generative abilities[36]. The dataset provides a learning foundation for the model. LLMs typically adopt a generative Transformer architecture [44], and in the fine-tuning stage [34], through the prompting method [1, 17, 43], the model can be fine-tuned according to task requirements to make it more suitable for specific tasks and scenarios. The Chain of Thought (CoT)[38] method is a foundational approach in prompting, which involves appending multiple reasoning steps before providing the answer to a question. This simple few-shot prompting strategy[4] has significantly enhanced the performance of Large Language Models (LLMs) in complex reasoning tasks [20]. Few-shot prompting [4] is effective across various tasks and has notably improved mathematical reasoning tasks. Extensions of CoT [11, 31, 32, 37] have further expanded the range of reasoning tasks that LLMs can tackle, improving their performance on various benchmarks. However, previous approaches have struggled with accuracy in arithmetic calculations and reasoning errors [12, 19, 26, 27]. To address complex calculations and reasoning, advanced prompt strategies like Program of Thought [7], Program-Aided Language (PAL) [35], MathPrompter [13], Least-to-Most Prompting [46], and Plan-and-Solve [35] have been developed. Prompt engineering has evolved

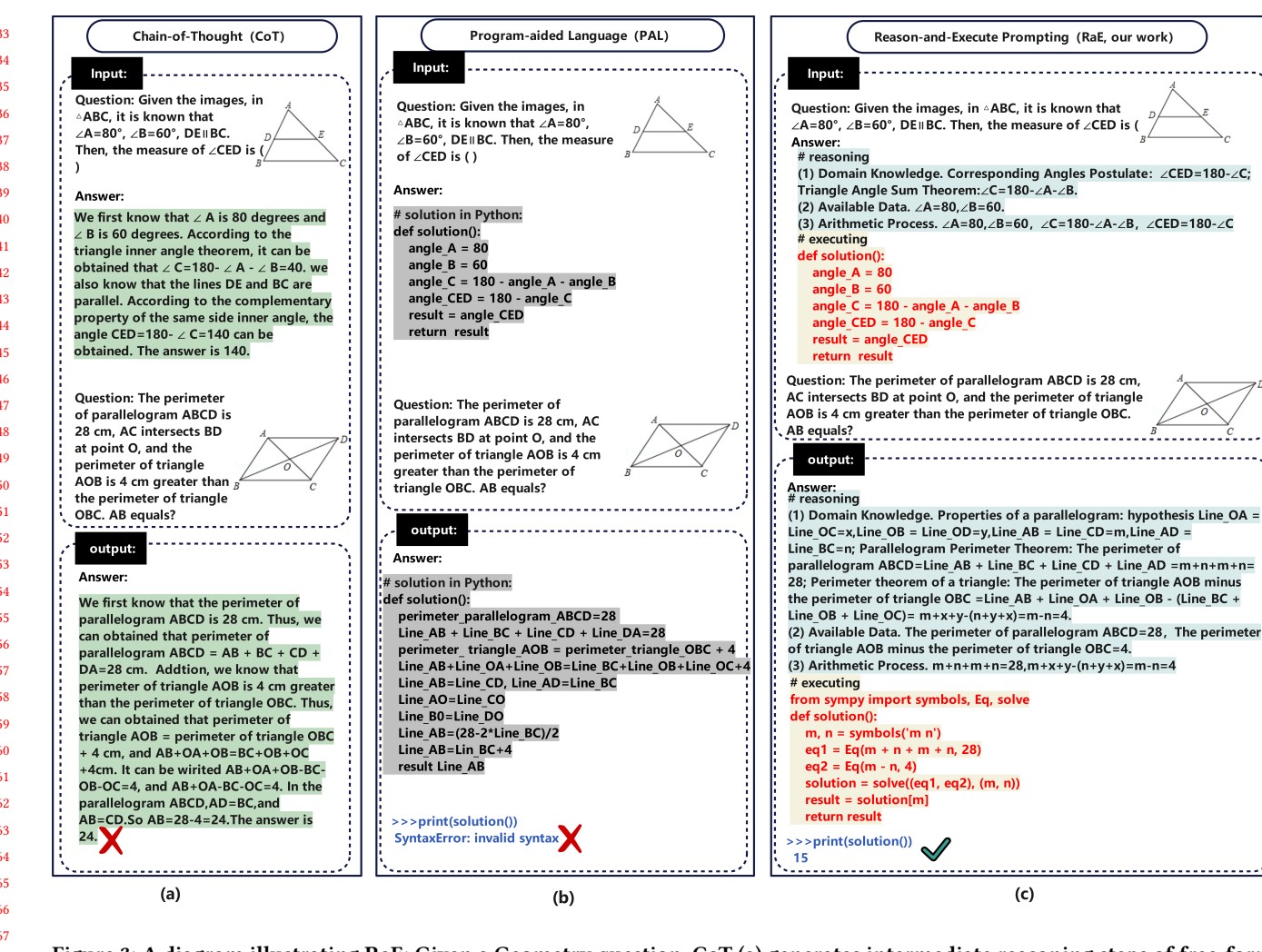

Figure 3: A diagram illustrating RaE: Given a Geometry question, CoT (a) generates intermediate reasoning steps of free-form text. Additionally, PAL (b) generates the Python code and shifts the role of running the reasoning steps from the language model to the Python interpreter. In contrast, our work, RaE (c), generates both rigorous reasoning processes and precise execution programs. The final answer is obtained by running the generated reasoning chain. CoT reasoning is highlighted in green; PAL is highlighted in gray; RaE is highlighted in orange and beige; the Python interpreter run is highlighted in blue.

from static to dynamic strategies such as Active Prompt, RLPrompt, and PRewrite[16]. Additionally, LLMs have demonstrated effectiveness in providing automated error feedback, highlighting their utility in solving math word problems. Among numerous prompt methods, CoT and PAL are pioneering methods for reason and algorithm, respectively, and are also the most widely applicable methods. Therefore, our subsequent research will expand these two methods into prompt templates suitable for multimodal tasks as our baseline prompt methods.

## 3 REASON-AND-EXECUTE PROMPTING

**Overview.** We introduce RaE prompting, a new prompting method specifically designed for MM-LLMs to solve geometric questions $q$, as shown in Figure 3. It ensures MM-LLMs generate reasoning processes $r$ by utilizing domain knowledge, and generate executable code blocks $b$ to obtain answers. Compared with the few-shot CoT and PAL methods, the RaE prompting method, designed for more

professional solving of geometric problems, features both rigorous reasoning processes and precise arithmetic operations. Specifically, RaE prompting leverages the advantages of MM-LLMs to solve tasks with a set of k examples, $\{(q_i, r_i, b_i)\}_{i=1}^k \parallel q_{test}$. Each example in the RaE prompt is a triplet of $< q_i, r_i, b1 >$, where $q_i$ and $b_i$ are input-output pairs, and $r_i$ is an inference process that ensures the solvability of code block $b_i$. Note that the test question $q_test$ we input does not directly generate an answer, but is obtained through the execution of the generated code block $b_test$ operation by the interpreter.

Overall, the RaE prompts involve two steps. In step 1, to solve geometric questions more professionally, we designed prompts for the inference process $r$ based on question-oriented thinking and domain knowledge, avoiding the interference of redundant information. In step 2, to obtain a precise arithmetic answer, a code block $b$ that can be executed by the interpreter is generated based

on the reasoning given in step 1, and the interpreter finally outputs the answer to the problem through the operation.

## 3.1 Step 1: Prompting for rigorous reasoning

To solve geometric questions more professionally, we adopted a Reverse thinking [9] approach, guiding MM-LLMs to start from the question being solved and gradually identify the known conditions necessary to solve the question, as shown in Figure 4. The template constructed in this step needs to meet the following three conditions:

```
# reasoning
(1) Domain Knowledge. Corresponding Angles Postulate:
∠CED=180-∠C; Triangle Angle Sum Theorem:∠C=180-∠A-∠B.
(2) Available Data. ∠A=80,∠B=60.
(3) Arithmetic Process. ∠A=80,∠B=60, ∠C=180-∠A-∠B, ∠CED=180-
∠C
```

Figure 4: Prompting template for rigorous reasoning

**Clarify domain knowledge** This template needs to analyze which theorems, formulas, properties, and other domain knowledge are needed to solve the geometric question and describe this domain knowledge in the form of equations combined with the information of the question. For example, as shown in the figure, to obtain the degree of $\angle CED$, one can use the complementary property of the *same side inner angles of parallel lines* to obtain $\angle CED = 180 - \angle C$; Although the degree of $\angle C$ here is not a known condition, it can be obtained through *the triangle angle sum theorem* that $\angle C = 180 - \angle A - \angle B$. This example involves two domain knowledge: *the property of complementary inner angles on the same side of parallel lines* and *the triangle angle sum theorem.*

**Clarify available data** The template needs to identify which data is needed to solve the question and obtain it from the graphic and textual content of the problem. For example, as shown in the figure, after analyzing domain knowledge, solving the problem requires the degrees of $\angle A=80$ and $\angle B = 60$. Note that sometimes the required data is not in the text of the question and needs to be identified from the image of the question.

**Clarify the arithmetic process** This template needs to integrate the domain knowledge and available data used in the question and clarify the operational process that needs to be transformed into program blocks. For example, as shown in this image, based on the analysis above, the process of performing precise arithmetic to answer this geometric question is: $\angle A = 80, \angle B = 60, \angle C = 180 - \angle A - \angle B, \angle CED = 180 - \angle C$.

In summary, our reasoning prompting template is a process that starts from the question to obtain the required data and clarifies the need for precise arithmetic.

## 3.2 Step 2: Prompting for precise arithmetic

It is easy to make mistakes when MM-LLMs rely solely on "memory" to obtain answers to the questions. To obtain precise answers to geometric questions, we use the idea of program-assisted problem-solving to guide MM-LLMs to understand the need for precise arithmetic from step 1 and generate executable code blocks, as shown in Figure 5. Finally, the precise answer is obtained by running the code block through the interpreter. The template constructed in this step must meet two conditions:

```
# executing
def solution():
    angle_A = 80
    angle_B = 60
    angle_C = 180 - angle_A - angle_B
    angle_CED = 180 - angle_C
    result = angle_CED
    return  result
```

Figure 5: Prompting template for precise arithmetic

**Unified block naming** The template needs to have a unified naming of code blocks to ensure that it can detect the generated executable programs. To obtain the final precise answer to a question, we must stably execute code blocks through an interpreter, and a unified naming of program blocks can enable the model to smoothly pass the interpreter's compilation. For example, as shown in the figure, we named the code block *"def solution"*.

**Meaningful variable naming** This template requires meaningful variable names to ensure that the program block has high runtime quality. Meaningful variable naming can to some extent avoid the problem of invalid program syntax. This is also related to whether the parameters involved in the arithmetic process in step 1 are clear. For example, as shown in the figure, based on the required parameters in step 1, we have designed variable names: *angle_A, angle_B, angle_C, angle_CED*.

Overall, to obtain an accurate answer to this geometric question, we must rely on the interpreter to smoothly execute the generated code block.

## 4 EXPERIMENTAL SETUP

### 4.1 Benchmarks

The proposed method is evaluated on the six benchmark datasets, as shown in Table 1. Geometry question datasets: (1) the **GEOS** [33]dataset contains simple middle school geometry problems with geometric shapes, (2) the **Geometry3K** [23] dataset contains numerous geometry questions where semantic information is scarce and most values need to be obtained from images, (3) the **GeoQA** [6] dataset contains rich semantic information for middle and high school geometry questions, (4) the **GeoQA+** [5] dataset is based on GeoQA, which adds more diverse types of geometry questions and forms an enhanced benchmark dataset. Other science question datasets: (1) the **AI2D** [14] dataset includes diagram questions for multiple natural science courses of the elementary school; (2) the **TQA** [15] dataset is drawn from middle school science curricula textbooks. The above datasets are all applicable and publicly available datasets for our work.

Figure 6 illustrates the statistical distribution of question length in the six benchmark datasets. In the GEOS (a), GeoQA (c), GeoQA (d), AI2D (e), and TQA (f) datasets, the distribution aligns with expected patterns, with the majority of questions containing substantial textual content. Moreover, the textual content appears to reasonably correspond to the information depicted in the diagram. Conversely, in the Geometry3K dataset (b), approximately 18% of all questions contain 3 words or fewer. This indicates a lack of descriptive information from the text of the question and primarily provides specific queries, such as *'Find UT'*.

**Table 1: Details of datasets being evaluated. The "total" represents the question number of questions in an original dataset, the "sample" represents the number of questions randomly selected from a dataset in a test, "en" represents the questions in English, and the "zh" represents the questions in Chinese.**

| Dataset | Total | Sample | Avg. words | Avg. knowledges | Domain | Level | Lange |
|---------|-------|--------|-----------|-----------------|--------|-------|-------|
| GEOS | 186 | 62 | 24.7 | 1.3 | Geometry | Middle school | en |
| Geometry3K | 3002 | 1000 | 12.2 | 1.6 | Geometry | Middle/High school | en |
| GeoQA | 4998 | 1666 | 52.5 | 2.1 | Geometry | Middle/High school | zh |
| GeoQA+ | 7528 | 2510 | 54.5 | 1.8 | Geometry | Middle/High school | zh |
| AI2D | 4908 | 1636 | 11.8 | 1.0 | Science | Elementary school | en |
| TQA | 15154 | 5051 | 9.8 | 1.4 | Science | Middle school | en |

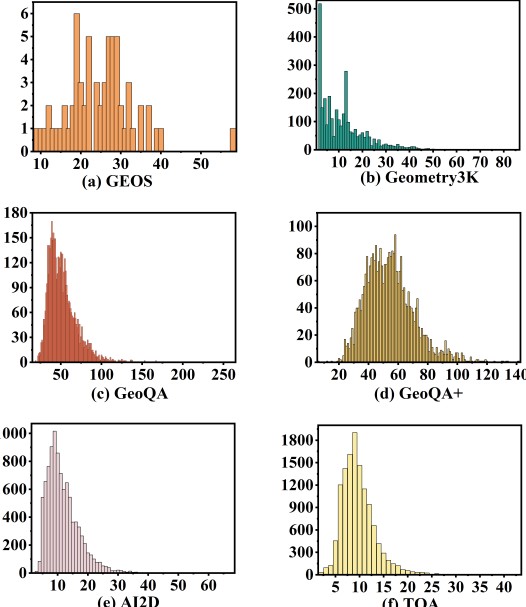

**Figure 6: Question length distribution of six benchmark datasets. The horizontal axis represents the number of question words, and the vertical axis represents the number of questions.**

### 4.2 Baselines

Since geometry questions are mostly multimodal, we adopted seven multimodal large language models as the baseline: (1) the **GPT-4V** [29] is a visually functional GPT-4; (2) the **Gemini-Pro** [18], as an upgraded version of Bard, can understand and combine information from different modalities; (3) the **Qwen-VL-Plus** [2] surpasses GPT-4V and Gemini in Chinese question-answering and text comprehension tasks; (4) the **Qwen-VL-Chat** [2] is a visual AI assistant based on a large language model, built based on Qwen-VL. (5) the **CogVLM** [35] model differs from the previous approach of only mapping visual features to the language input space by adding a visual expert module at each Transformer layer; (6) the **InternLM XComponent** [8] is a visual language model that features interleaved text image combinations and multilingual knowledge-based understanding; (7) the **mPLUG-Owl2** [2] is the first MLLM model to demonstrate modal collaboration phenomena in both pure text and multimodal scenarios; (8) the **Deep-seek** [21] is a

model that emphasizes data diversity, model efficiency, and balance; (9) **Yi-VL**[40] is a model developed based on the Yi language model, suitable for massive data analysis, mining, and cross-domain knowledge fusion. The above MM-LLMs can effectively understand and process multiple languages and visual information, achieving more accurate and comprehensive question-answering and text understanding.

### 4.3 Implementations

We evaluate the performance of various MM-LLMs on the six benchmarks, including both closed-source and open-source models. The closed-source models are evaluated by using their official API, while open-source models are evaluated by running inferences on 4-way RTX 4090GPU. For the closed-source models, we select state-of-the-art models **GPT-4V (gpt-4-vision-preview)**, **Gemini-Pro (gemini-pro-vision)**, and **Qwen-VL-Plus (qwen-vl-plus)**. For the open-source models, model sizes vary from 6b to 7b, including **CogVLM(cogvlm-7b)**, **Qwen-VL-Chat (qwen-vl-chat-7b)**, **Intern-XCompose (intern-xcomposer-7b)**, **Yi-VL(yi-vl-6B)**, **DeepSeek-VL(deep-seek-vl-7b)**, and **MPLUG-Owl2(mplug-owl2-7b)**. The templates used in the experiment can be found in "Appendix A". In addition, We performed greedy decoding from the language model using a temperature of 0. Meanwhile, considering the real-world usage of the model, we simulated the use of different MM LLMs by users: randomly selecting the sample size shown in Table 1 from each dataset for testing. A total of 10 rounds were selected, and the average accuracy was taken as the accuracy of the final answer.

## 5 EXPERIMENTAL RESULTS

### 5.1 Main Result

The experimental results of applying various prompt methods to solve geometry and science questions in different MM-LLMs are shown in Table 2. Due to the weak code generation ability of open-source models, we only used closed-source models when using PAL and RaE, while CoT and silent methods used all baseline models. Experimental results show that RaE outperforms all other prompting methods across the geometry datasets of GEOS (32.6%), Geometry3k (32.3%), GeoQA (31.1%), and GeoQA+ (29.3%). However, it performs less effectively than the COT method on the science datasets, A12D (79.6%) and the TQA (73.4%), indicating that our method significantly improves the accuracy of MM-LLMS in answering geometry questions.

**Table 2: Answer accuracy comparison on the six benchmark datasets.**

| Setting | Model | Geometry | | | | Science | |
|---|---|---|---|---|---|---|---|
| | | GEOS | Geometry3K | GeoQA | GeoQA+ | A12D | TQA |
| Without prompting | gpt-4-vision-preview | 19.8 | 20.2 | 25.2 | 26.5 | 78.2 | 71.0 |
| | gemini-pro-vision | 9.0 | 11.4 | 17.9 | 14.7 | 73.9 | 73.0 |
| | qwen-vl-chat-7b | 11.2 | 4.0 | 13.1 | 9.5 | 70.1 | 49.4 |
| | cogvlm-7b | 7.4 | 3.2 | 9.5 | 6.4 | 56.2 | 39.7 |
| | qwen-vl-plus | 13.6 | 9.2 | 16.7 | 14.8 | 75.9 | 69.5 |
| | intern-xcomposer-7b | 9.1 | 3.7 | 15.5 | 12.3 | 30.9 | 20.4 |
| | mplug-owl2-7b. | 8.4 | 2.8 | 8.9 | 5.6 | 27.1 | 18.7 |
| | yi-vl-6B | 9.5 | 3.6 | 10.1 | 8.3 | 64.7 | 56.3 |
| | deep-seek-vl-7b | 8.6 | 3.1 | 9.0 | 7.4 | 56.2 | 47.0 |
| CoT prompting | $CoT_{(gpt-4-vision-preview)}$ | 29.5 | 27.2 | 28.6 | 28.2 | **80.1** | **74.5** |
| | $CT_{(gemini-pro-vision)}$ | 16.4 | 4.7 | 15.4 | 17.0 | 76.2 | 75.7 |
| | $CoT_{(qwen-vl-chat-7b)}$ | 4.4 | 3.0 | 8.5 | 7.7 | 71.3 | 51.5 |
| | $CoT_{(cogvlm-7b)}$ | 2.8 | 0.9 | 4.7 | 5.1 | 58.2 | 49.1 |
| | $CoT_{(qwen-vl-plus)}$ | 5.4 | 2.2 | 15.3 | 12.4 | 79.7 | 75.3 |
| | $CoT_{(intern-xcomposer-7b)}$ | 10.2 | 2.3 | 12.4 | 12.3 | 31.2 | 22.1 |
| | $CoT_{(mplug-owl2-7b)}$ | 5.1 | 8.3 | 8.3 | 6.1 | 29.9 | 20.3 |
| | $CoT_{(yi-vl-6B)}$ | 7.2 | 3.7 | 9.4 | 8.9 | 67.2 | 60.3 |
| | $CoT_{(deep-seek-vl-7b)}$ | 5.6 | 2.1 | 6.7 | 7.2 | 59.8 | 48.4 |
| PAL prompting | $PAL_{(gpt-4-vision-preview)}$ | 26.3 | 25.0 | 27.7 | 25.3 | 50.1 | 43.7 |
| | $PAL_{(gemini-pro-vision)}$ | 12.4 | 2.7 | 8.9 | 9.1 | 46.9 | 42.7 |
| | $PAL_{(qwen-vl-plus)}$ | 8.4 | 3.7 | 7.1 | 6.8 | 44.5 | 37.3 |
| RaE prompting (our work) | $RaE_{(gpt-4-vision-preview)}$ | **32.6** | **32.3** | **31.1** | **29.3** | 79.6 | 73.4 |
| | $RaE_{(gemini-pro-vision)}$ | 15.7 | 5.1 | 17.9 | 19.8 | 72.8 | 71.1 |
| | $RaE_{(qwen-vl-plus)}$ | 10.4 | 3.0 | 11.8 | 9.5 | 74.3 | 68.2 |

From Table 2, we can also observe that the performance of PAL is relatively poor in these six datasets. This is because PAL generates code blocks instead of reasoning through natural language. However, when solving geometry questions, the "def solution()" generated code block contains reasoning steps, rendering the entire code block inoperable. Our method separates the reasoning of geometry questions from the code generation process and uses the generated rigorous reasoning process to guide MM-LLMs in generating executable code blocks, thereby achieving precise arithmetic. In summary, our proposed RaE is more suitable for solving geometric problems in closed-source MM-LLMs than other prompt methods and also enhances the ability of MM-LLMs to solve geometry questions.

## 5.2 Analysis

Solving geometry questions is a comprehensive test of the various abilities of the MM-LLMs, especially for our proposed RaE prompting method. For this, we have considered multiple factors from the perspective of problem-solving, including domain knowledge, geometry shapes, understanding of the question text, and language use. We tested the answer performance of GPT-4V models with RaE, PAL, and CoT prompting methods, and without prompting methods on these factors. Below are specific experimental analyses based on GPT-4V.

**Which is most important for RaE prompt templates, reasoning or executing?** Our work proposes a prompting method

**Table 3: Ablation experiments of the RaE prompting.**

| Datasets | GEOS | Geometry3K | GeoQA | GeoQA+ |
|---|---|---|---|---|
| RaE w/o Reasoning | 25.9 | 24.7 | 27.2 | 25.0 |
| RaE w/o Executing | 28.4 | 29.6 | 30.1 | 27.8 |
| RaE | **32.6** | **32.3** | **31.1** | **29.3** |

called RaE, which mainly contributes to the design of a prompting template with rigorous reasoning and precise arithmetic parts. To explore the importance of these two parts for RaE prompts, we conducted ablation experiments using the same experimental setup as the Main Result. From the data in Table 3, it can be seen that the rigorous reasoning part has a greater impact on the performance of RaE. Specifically, the lack of an inference section resulted in a maximum decrease of 7.6% in RaE's accuracy in solving geometric problems (on the Geometry3K dataset), while the maximum decrease without an execution section was only 4.2% (on the GEOS dataset). This indicates that using professional domain knowledge to guide MM LLMs in answering questions can fundamentally improve performance while using program solving can assist in improving limited performance.

**Does RaE work with Multi-domain knowledge?** In the previous analysis, we found that rigorous reasoning is crucial in the RaE template. To achieve this rigorous reasoning, one needs to consider the domain knowledge required to solve the question. Therefore,

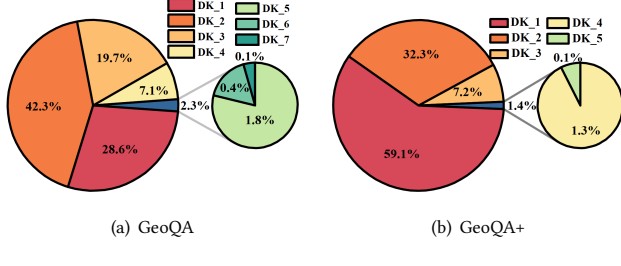

(a) GeoQA

(b) GeoQA+

**Figure 7: The distribution of the number of questions involving knowledge from different domains in two datasets, GeoQA and GeoQA+.** $DK\_i$ indicates that answering a geometry question requires at least $i$ domain knowledge.

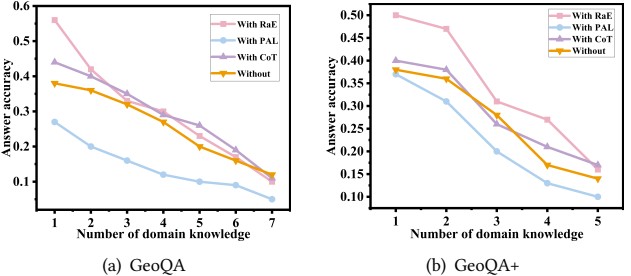

(a) GeoQA

(b) GeoQA+

**Figure 8: The accuracy of answering geometric problems with varying numbers of domain knowledge. 'With RaE', 'With CoT', 'With PAL', and 'Without' respectively represent GPT4V models with RAE prompting, CoT prompting, PAL prompting, and no prompting.**

we analyzed the domain knowledge involved in the GeoQA and GeoQA+ datasets and counted the number of questions. The detailed domain-knowledge statistics can be found in "Appendix B". In these statistics, we found that solving a large number of questions relies on mixed domain knowledge. That is, solving a geometric question may require mastering two domain knowledge points, or it may require mastering $i$ domain knowledge. Here, $i$ is taken as 1 to 7 in the GeoQA dataset and 1 to 5 in the GeoQA+dataset. As shown in Figure 7, Approximately 42.3% of geometric problems can only be solved by combining two domain knowledge and 40.9% of the questions in GEOQA+ also contain more than one domain knowledge. To further analyze the impact of domain knowledge on solving geometry questions, we randomly selected 5 questions for each type of domain knowledge quantity, for a total of 100 rounds. The experimental results are shown in Figure 8, when the number of domain knowledge increases, the accuracy of all methods decreases. According to Figure 8 (a), RaE performance remains high when solving questions involving less domain knowledge. PAL's performance is the worst. According to Figure 8 (b), when the number of domain knowledge increases, its performance is always superior to other prompting methods. The results show that RaE is more suitable than other methods for solving geometry questions with multi-domain knowledge.

**Is the error source of RaE prompting templates the reasoning or executing?** Although our prompting method RaE performs

**Table 4: Statistics on the sources of problem-solving errors.** $R_e$ **represents code execution error,** $R_r$ **represents reasoning process error**

| Model | GEOS | | Geometry3K | | GeoQA | | GeoQA+ | |
| --- | --- | --- | --- | --- | --- | --- | --- | --- |
| | $R_r$ | $R_e$ | $R_r$ | $R_e$ | $R_r$ | $R_e$ | $R_r$ | $R_e$ |
| PAL | 28.6 | 45.1 | 35.2 | 40.0 | 17.8 | 54.5 | 13.4 | 61.2 |
| RaE | 35.7 | 32.2 | 38.3 | 29.4 | 37.9 | 30.7 | 32.1 | 38.5 |

well compared to other prompt methods in solving geometry questions, there is still room for improvement. Therefore, we analyzed the probabilities of reasoning errors and code execution errors using the same experimental setup as the Main Result in four geometry datasets. As shown in Figure 3, we consider the program output " SyntaxError: invalid syntax " as a code execution error ($R_e$), and consider the generated answer not being numerically equal to the true answer as a reasoning process error ($R_r$). Since errors in COT prompt methods are all caused by $R_r$, we do not compare them here. From Table 4, it can be analyzed that the main reason for PAL's error in answering questions is due to code execution errors. The error probability of our proposed RaE method in code execution is much lower than that of the PAL prompt method. Therefore, for PAL, to enhance the performance of solving geometric questions, it is necessary to optimize the generation of code blocks. For our work, we need to provide a more concise and rigorous reasoning process to guide MM-LLMs to achieve professional solutions.

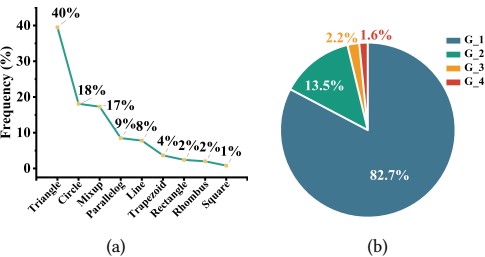

(a)

(b)

**Figure 9: Geometry shape distribution of Geometry3K. (a) Geometry shape distribution statistics, (b) Statistics of the number distribution of geometry shapes,** $G\_i$ **indicates that a geometry question includes at least** $i$ **Geometry shape.**

**How does RaE work with different geometry shapes?** The most important thing in solving geometry questions is to recognize geometric shapes. To analyze the impact of different geometry shapes on MM-LLMs with different prompting methods, we analyzed the distribution of questions with different shapes in the geometry3k dataset. The analysis results are shown in Figure 9, according to Figure 9 (a), questions containing only triangles account for 40% of the total number, and 17% of the questions contain more than one shape. Furthermore, as shown in Figure 9 (b), more than 82% of the questions in the dataset geometry3k contain only one kind of shape. To further analyze the impact of geometry shapes on solving geometry questions, we randomly selected 5 questions for each type of geometry shape, for a total of 100 rounds. The experimental results are shown in Figure 10. The experimental results are

explicit, Our RaE has the best performance in all kinds of geometric shapes, especially in quadrangles and triangles. The performance of CoT is only inferior to RaE, and the gap between CoT and RaE is the largest in the triangle. PAL and no prompt were the worst. To further analyze the number impact of geometry shapes on solving geometry questions, we randomly selected 40 questions for each type of geometry shape quantity, for a total of 50 rounds. The experimental results are shown in Figure 11. According to Figure 11, our RaE is the best in solving questions with one to three kinds of geometry shapes. When the number of shapes increases to four, COT performs better than all other prompting methods. This is caused by the fact that the reasoning process of RaE's prompt template did not fully consider the questions of mixing multiple shapes.

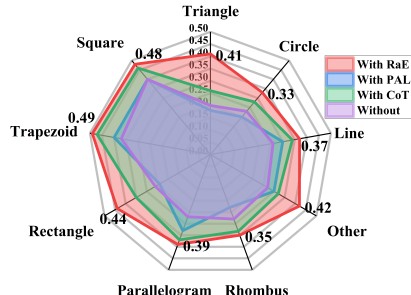

Figure 10: Accuracy of answering with different geometry shapes

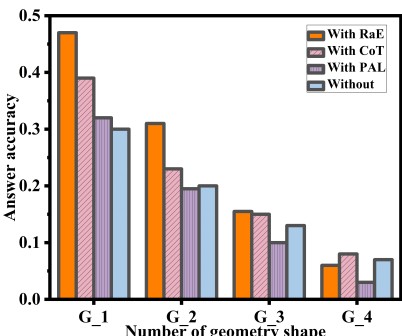

Figure 11: Accuracy of answering with the different number of geometry shapes. $G\_i$ indicates that a geometry question includes at least $i$ Geometry shape.

**What length of question text is suitable for MM-LLMs to solve under RaE prompts?** In addition to mastering relevant domain knowledge and understanding geometric shapes, it is more important to understand the meaning of exercises when solving geometric problems. To analyze the understanding of exercise questions by MM LLMs, we randomly selected one question from each length of the GeoQA dataset and conducted a total of 100 rounds to test the accuracy of MM-LLMs in answering questions of different lengths. The test results are shown in Figure 12. When the number of question words is around 35 to 57, the accuracy of MM LLMs in answering is at a high level. This indicates that when asking GPT4V, we should try to keep it within 60 words. The model can provide a more accurate answer.

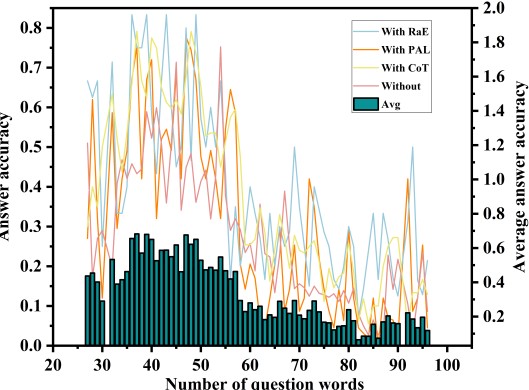

Figure 12: Accuracy of answering with the different number of question words, in the GeoQA dataset. The Avg refers to the average answering accuracy of GPT4V with RaE, PAL, CoT prompting method, and no prompting method.

**Is RaE more suitable for "zh" or "en"?** Due to the presence of both English and Chinese questions in our original geometry datasets, to minimize the interference factors of MM-LLMs in answering the questions, we unified the language of the four geometric data. Thus, we analyzed the impact of the language used in the question on the accuracy of the answer using the same experimental setup as the Main Result in four geometry datasets. The experimental results are shown in Table 5. The results demonstrate that our RaE and other prompting methods are more suitable for English questions, and our answering accuracy reaches the highest of 32.7% on the GeoA_en dataset.

Table 5: The accuracy statistics of different prompting methods for geometry questions in different languages.

| Model | GEOS | | Geometry3K | | GeoQA | | GeoQA+ | |
|---|---|---|---|---|---|---|---|---|
| | zh | en | zh | en | zh | en | zh | en |
| CoT | 29.5 | 26.1 | 27.2 | 26.7 | 31.2 | 28.6 | 30.4 | 28.2 |
| PAL | 26.3 | 25.7 | 25.0 | 23.3 | 29.1 | 27.7 | 26.5 | 25.3 |
| RaE | 32.6 | 29.4 | 32.3 | 28.7 | 32.7 | 31.1 | 32.0 | 29.3 |

## 6 CONCLUSION

In this paper, We introduce RaE prompting, a new prompting method specifically designed for MM-LLMs to solve geometric questions. It ensures MM-LLMs generate reasoning processes by utilizing domain knowledge, and generate executable code blocks to obtain answers. Compared with the few-shot CoT and PAL methods, the RaE prompting method, designed for more professional solving of geometric problems, features both rigorous reasoning processes and precise arithmetic operations. From the overall results of the experiment, our RaE showed impressive performance on four geometry question datasets. To analyze the influencing factors of solving geometry questions in more detail, we tested the answering performance of different prompting methods based on the GPT4V model, demonstrating rich experimental results. Our work provides a more comprehensive research approach to improving large language models for solving geometry questions and points the way for future research.

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
