# OpenReview forum: "Reason-and-Execute Prompting: Enhancing MultiModal Large Language Models for Solving Geometry Questions"
_acmmm.org/ACMMM/2024/Conference — MM2024 Poster_

### Official Review · Reviewer_QAu2 · 2024-05-20

**Rating:** 4
**Confidence:** 3

**Summary:**

The work proposes RaE, a novel prompting method called Reason-and-Execute (RaE). The RaE is designed to improve the performance of MultiModal Large Language Models in solving geometry questions.

This approach involves two main steps:
- [A] Designing a rigorous reasoning process based on domain knowledge
- [B] Generating executable code blocks for precise arithmetic.

The authors evaluate their method against existing prompting techniques like Chain-of-Thought (CoT) and Program-Aided Language (PAL) across various datasets, demonstrating that RaE significantly enhances accuracy in solving geometric problems.

**Strengths:**

- Approach wise: The RaE prompting method introduces an innovative way to enhance MM-LLMs' ability to solve geometry problems by combining domain-specific reasoning with precise arithmetic execution.
- Evaluation and analysis: The paper includes thorough experimental evaluations on six benchmark datasets, showcasing the effectiveness of RaE compared to CoT and PAL prompting methods. The authors also provide an in-depth analysis of error sources, distinguishing between reasoning and execution errors, which offers valuable insights for further improvements.
- Scalability Considerations: The paper acknowledges the computational demands and addresses the scalability of their approach by evaluating both closed-source and open-source models.

**Limitations:**

As for baselines, while the paper compares RaE with CoT and PAL, it lacks a discussion on why certain other relevant prompting methods were not included in the evaluation.

I also have a few questions.

- Scalability: What specific optimizations were employed to manage the computational load (e.g. LoRA), and are there any plans to improve the scalability of the RaE method for larger datasets?
- Practical Application: How does the proposed framework translate to real-world applications, particularly in educational contexts? Are there any ongoing pilot projects or collaborations with educational institutions?
- Feature Integration: How does the model handle potential conflicts or redundancies when integrating multiple domain-specific knowledge points in the reasoning process?

**Suitability:**

2

---

### Official Review · Reviewer_BwT7 · 2024-05-21

**Rating:** 6
**Confidence:** 3

**Summary:**

This paper demonstrates the existing challenges that multi-modal large language models face in answering geometric questions, through an error analysis conducted on GPT-4V. Then, it proposes a Reason-and-Execute prompting method to enhance multi-modal LLMs to solve geometric questions. Experiments on six benchmarks show the effectiveness of its methods.

**Strengths:**

- This paper presents an interesting and meaningful problem: how to enhance multimodal large language models to better handle geometric questions. It also exposes the potential challenges through an error analysis conducted on GPT-4V, clearly articulating the motivation behind this work.

- This paper is well-written, and the figures are clear and easy to understand.

- The method proposed in this paper is simple and intuitive, yet appears to be reasonable.

- This paper conducts experiments on 6 benchmarks and adopts the latest multimodal large language models (e.g. GPT-4V, Gemini, and Qwen). The experimental results demonstrate significant improvements and quantitative analyses showcase the effectiveness of the proposed method.

**Limitations:**

The paper does not have any obvious shortcomings, but I still have the following concerns:
- Is the method proposed in this paper sensitive to the choice of prompts? The authors should provide some guidance on how to select prompts. Additionally, the authors should release their code and prompts after publication to ensure reproducibility.

- There are some related works that solve mathematical reasoning problems through code generation. The authors should mention these papers in the related work section, such as [1].

[1] PaD: Program-aided Distillation Can Teach Small Models Reasoning Better than Chain-of-thought Fine-tuning.

**Suitability:**

3

---

### Official Review · Reviewer_rfwa · 2024-05-24

**Rating:** 4
**Confidence:** 2

**Summary:**

By combining domain knowledge and program-aid strategies in prompts with a method called RaE, this article alleviates the problem of generating unexecutable code in the PAL method and promotes the ability of multi-modal large models to solve geometric problems. RaE is evaluated on 9 models across 6 datasets, which nicely demonstrates the broad applicability of the method.

Overall, RaE is a simple but efficient prompt strategy to stimulate the performance of MM-LLMs.

**Strengths:**

1. RaE is evaluated on 9 models across 6 datasets, which nicely demonstrates the broad applicability of the method.
2. RaE is simple but efficient.
3. This paper is expressed with clarity and fluidity.

**Limitations:**

1. RaE is not as effective as CoT on some models and data sets (GeoQA+ and qwen-vl-plus, etc.), but the author does not provide an analysis.
2. The experimental part does not show case studies (including good case and bad case).
3. It is necessary to compare some methods of improving PAL, such as PoT[1]
4. It is recommended that the presentation form of Table 2 be adjusted so that the performance of different methods on the same model and dataset can be clearly compared.
5. Since LLMs are prone to hallucination when generating specific domain knowledge, it is recommended to evaluate the accuracy of the following "#reasoning" part separately.

**Suitability:**

3

---

### Meta-Review · Area_Chair_vd4K · 2024-07-03

**Recommendation:** Accept (Poster)
**Confidence:** 5

**Metareview:**

This is a good submission and all reviewers are quite positive. All of the concerns have been addressed in the rebuttal so reviewers voted for an acceptance. The final decision is accept.